# Effect of Host-Associated *Bacillus*-Supplemented Artificial Diets on Growth, Survival Rate, and Gene Expression in Early-Stage Eel Larvae (*Anguilla japonica*)

**Won Je Jang** [1,2,†] , **Shin-Kwon Kim** [3,†] , **So Young Park** [1] , **Dong Pil Kim** [1] , **Yun-Jy Heo** [1] , **Haham Kim** [4] ,
**Su-Jeong Lee** [2] , **Min Gyu Shin** [3] , **Eun-Woo Lee** [2] , **Seunghyung Lee** [4] and **Jong Min Lee** [1,*]

1 Department of Biotechnology, Pukyong National University, Busan 48513, Republic of Korea; jangwj9914@pukyong.ac.kr or wjjang@deu.ac.kr (W.J.J.); thdud4533@pukyong.ac.kr (S.Y.P.); dkimp7994@pukyong.ac.kr (D.P.K.); yunjy1115@pukyong.ac.kr (Y.-J.H.)
2 Biopharmaceutical Engineering Major, Division of Applied Bioengineering, Dong-Eui University, Busan 47340, Republic of Korea; 14480@deu.ac.kr (S.-J.L.); ewlee@deu.ac.kr (E.-W.L.)
3 Aquaculture Research Division, National Institute of Fisheries Science, Busan 46083, Republic of Korea; ksk4116@korea.kr (S.-K.K.); smg159@korea.kr (M.G.S.)
4 Major of Aquaculture and Applied Life Sciences, Division of Fisheries Life Sciences, Pukyong National University, Busan 48513, Republic of Korea; haham7@pukyong.ac.kr (H.K.); shlee@pknu.ac.kr (S.L.)
* Correspondence: jmlee84@pknu.ac.kr; Tel.: +82-51-629-5865
† These authors contributed equally to this work.

**Abstract:** Beneficial microorganisms can increase nutrient digestion and absorption in farmed fish. This study investigates the effects of supplemental feeding of *Bacillus* species isolated from the intestines of wild glass eels on the growth, survival, and gene expression of farm-raised eel larvae for 30 days after hatching. Three species of *Bacillus* (*B. velezensis*, AJBV; *B. subtilis*, AJBS; *B. licheniformis*, AJBL) without hemolytic activity were isolated, and an experiment compared the growth of eel larvae fed an artificial diet supplemented with each *Bacillus* species. There were no significant differences in the total length and body depth of eel larvae at 30 days after hatching in all groups. During the feeding period, 149 eels survived from the initial 1000 in the control group. On the other hand, 240, 178, and 141 eels survived in the AJBV, AJBS, and AJBL groups, respectively. However, there were no significant differences in survival rates despite the difference in the number of surviving eel larvae among the groups. In the comparison of gene expression of genes involved with growth (growth hormone, growth hormone receptor 1, insulin-like growth factor II-2) and those involved with digestive enzymes (amylase, trypsin, lipase), there were also no significant differences among the groups. Our results confirm that dietary supplementation with each of the three host-associated *Bacillus* does not affect the growth and survival rates of eel larvae reared on an artificial diet up to the first 30 days after hatching, nor does it significantly affect related gene expression.

**Keywords:** *Anguilla japonica*; eel; probiotics; host-associated probiotics

**Key Contribution:** Dietary supplementation with each of the three host-associated *Bacillus* does not affect the growth and survival rates of eel larvae reared on an artificial diet up to the first 30 days after hatching, nor does it significantly affect related gene expression.

## 1. Introduction

The eel has long been esteemed as an important food fish not only in East Asian countries such as Japan, Korea, China, and Taiwan but also in European countries [1]. Among eels, the Japanese eel (*Anguilla japonica*) is one of the species farmed in East Asian countries [2,3]. To date, all the stocking for eel aquaculture is dependent on wild glass eels, naturally sourced juveniles of eel collected from estuaries [1,4]. However, due to overfishing,

environmental degradation, climate change, and other unknown factors, the natural stock of Japanese eels has noticeably declined over the past few decades [5]. This decrease creates an unstable supply of glass eels for seeding, which has led to price increases [1]. Therefore, artificial propagation technology should be developed to protect natural glass eel resources and stabilize the aquaculture industry through a stable supply of glass eels for farming, which in turn can stabilize eel prices [1,3].

Research on the production of artificial eel larvae started in Japan in the 1960s. Currently, active research is conducted not only in Japan but also in Korea and China [6,7]. However, only small-scale production is possible in the laboratory because a suitable rearing system and feed have not yet been found [8]. In addition, artificially produced eel larvae exhibit slower growth and lower survival rates than naturally sourced individuals [2,9]. Therefore, further research progress in the rearing system and feed development are needed to overcome these deficiencies.

Many studies on probiotics have shown consistent results in that they can help improve growth and immunity in various farmed fish species [10]. In addition, recently, it has also been reported that bacteria isolated from the intestines of the fish studied may be the best potential probiotics [11–13]. Later these were called host-associated probiotics (HAPs) by Van Doan et al. (2020), who defined them as "bacteria originally isolated from the rearing waters or gastrointestinal tract to the host to improve the growth and health of the host" [14]. HAPs may be highly functional because they can evade the host's defense system and are better adapted to the host's intestinal environment [13,14].

This study was conducted to verify the effect of host-associated microorganisms on the growth and survival rate of eel larvae reared on an artificial diet. Three species of isolated host-associated microorganisms were separately supplied to eel larvae for 30 days after hatching, and growth and survival rates and related gene expression were investigated.

## 2. Materials and Methods

### 2.1. Bacterial Isolation and Identification

Microorganisms were isolated from wild glass eels captured in the Geumgang River ($36°00'14.4''$ N $126°43'52.4''$ E). The glass eels were ground and suspended in 0.85% NaCl, then spread on MRS (Difco, Detroit, MI, USA), LB (Difco, Detroit, MI, USA), and MB (Difco, Detroit, MI, USA) agar plates and cultured at 37 °C for 24 h. Colonies cultured on agar plates were separated according to size, color, and shape and then identified by performing 16S rRNA sequence analysis using 27F (AGA GTT TGA TCM TGG CTC AG) and 1492R (GGT TAC CTT GTT ACG ACT T) 16S universal primers (Bionics, South Korea). A phylogenetic tree of microorganisms based on 16s rRNA sequences was constructed using the neighbor-joining algorithm of the Molecular Evolutionary Genetics Analysis version 7 (MEGA7) software. Hemolytic activity was determined by incubating the microorganisms on a 5% sheep's blood agar base plate (Kisan Bio, Seoul, Republic of Korea) and confirming the formation of a clear area around the colony.

### 2.2. Experimental Artificial Diet Preparation

Table 1 lists the ingredients for preparing the slurry-type feed used in the experiment. Each feed component was mixed and filtered at 90 μm before use [3]. Microorganisms were cultured at 37 °C in LB broth and then washed twice with phosphate-buffered saline and treated to a final concentration of $1 \times 10^{11}$ colony-forming units (CFU)/mL.

**Table 1.** Composition of the basal experimental diet.

| Ingredient | Content |
|---|---|
| Shark egg | 50 g |
| Krill meal | 6 g |
| Soybean peptide | 3 g |
| Fishmeal | 3 g |
| Vitamin Mix | 0.3 g |
| Host-associated *Bacillus* | $1 \times 10^{11}$ CFU/mL |

### 2.3. Rearing System and Condition for the Feeding Trial

Eel larvae were provided by the National Institute of Fisheries Science. The rearing system and environment were the same as above [3]. Briefly, a feeding experiment was conducted in a 20 L round acrylic resin tank accommodating 330 eel larvae (3 tank/group, triplicates). Water quality was regularly monitored, and stable environmental parameters were maintained. The water temperature was maintained at $23 \pm 0.1$ °C, and the flow rate was 0.99–1.13 L/min. Feed was provided five times (09:00, 11:00, 13:00, 15:00, and 17:00) daily at the bottom of each tank after stopping the flow of water.

### 2.4. Growth and Survival Rate

The growth and survival rates of eel larvae supplied with the experimental feed were calculated 30 days after hatching. Thirty eel larvae were randomly selected per tank, and the total length and body depth were measured under a stereomicroscope (Stereo Discovery V.20, Carl Zeiss, Jena, Germany). Deformed individuals were excluded. The survival rate was calculated using the following equation: Survival rate (%) = the number of surviving eel larvae/number of hatched eel larvae $\times$ 100.

### 2.5. Gene Expression Analysis

Total RNA was isolated by pooling five eel larvae per tank (3tank/group, triplicates). The pooled samples were ground using a disposable tissue grinder pestle (Thermo Fisher Scientific, Waltham, MA, USA), and total RNA was isolated using the Hybrid-R RNA Purification Kit (GeneAll Biotechnology, South Korea) according to the manufacturer's instructions. Residual DNA was removed using the Riboclear plus kit (GeneAll Biotechnology, Seoul, Republic of Korea), and the purity and concentration of isolated RNA were evaluated using a NanoDrop Lite Spectrophotometer (Thermo Fisher Scientific, Waltham, MA, USA). cDNA was synthesized using the PrimeScript 1st strand cDNA Synthesis Kit (Takara, Japan). RT-qPCR was performed using TB Green Premix Ex Taq (Takara, Kusatsu, Japan) on a CFX96 system (Bio-Rad, Hercules, CA, USA) at the Core-Facility Center, Dong-Eui University (Busan, Republic of Korea). Relative quantification was calculated using the $2^{-\Delta\Delta CT}$ method and normalized to the β-actin gene. Primers used for gene expression analysis are shown in Table 2.

**Table 2.** Gene-specific primers used to quantify relative gene expression.

| Gene | Sense | Sequence (5′-3′) | Size (bp) | Access No. |
|---|---|---|---|---|
| β-actin | F | TGT GGA TCA GCA AGC AGG AG | 110 | GU001950.1 |
| | R | CAG TTT TGA GTC GGC GTG TG | | |
| Growth hormone | F | TGC ACA AAG TGG AGA CCT ACC | 118 | M24066.1 |
| | R | TTA ACA CAG GAC CGA AGC CC | | |
| Growth hormone receptor 1 | F | TCG CTG TTG ACA ACT TTG CG | 147 | AB180476.1 |
| | R | ACA GGC AAG GGG TGA AGA TG | | |
| Insulin-like growth factors II-2 | F | ACC TGC GTA AGG ACA GCA AA | 157 | AB353117.1 |
| | R | CTG CTGGTG GGT CTG CTA AA | | |

**Table 2.** *Cont.*

| Gene | Sense | Sequence (5′-3′) | Size (bp) | Access No. |
|---|---|---|---|---|
| Amylase | F | GTG GAG AAC CCA TTA CGG CA | 154 | AB070721.1 |
|  | R | CCA TCC TTC ACC CCA GGT TC |  |  |
| Trypsin | F | CCG AGC TTC CAA GGT TCT CC | 137 | AB519643.1 |
|  | R | GGG TTC ATA GTG TTG CCC CA |  |  |
| Lipase | F | GGG TTT GCT GGG AGC TAC TT | 126 | EU715406.1 |
|  | R | GCG TCC ACA AAT ATG GCG TC |  |  |

*2.6. Statistical Analysis*

All data were analyzed by one-way analysis of variance (ANOVA) using IBM's Statistical Package for the Social Sciences software (SPSS Inc., version 17.0, Chicago, IL, USA) followed by Duncan's multiple range test. Statistical significance was accepted at a *p*-value < 0.05 unless otherwise noted.

## 3. Results and Discussion

*3.1. Bacterial Isolation and Identification*

Three species of *Bacillus* (*B. velezensis*, AJBV; *B. subtilis*, AJBS; *B. licheniformis*, AJBL) were identified through 16S rRNA sequence analysis. The three selected *Bacillus* species shared high similarities with *B. velezensis* CR-502[T], *B. subtilis* NCIB 3610[T], and *B. licheniformis* ATCC 14580[T], respectively (Figure 1a). Hemolytic activity was not observed in any of the isolated *Bacillus* strains (Figure 1b).

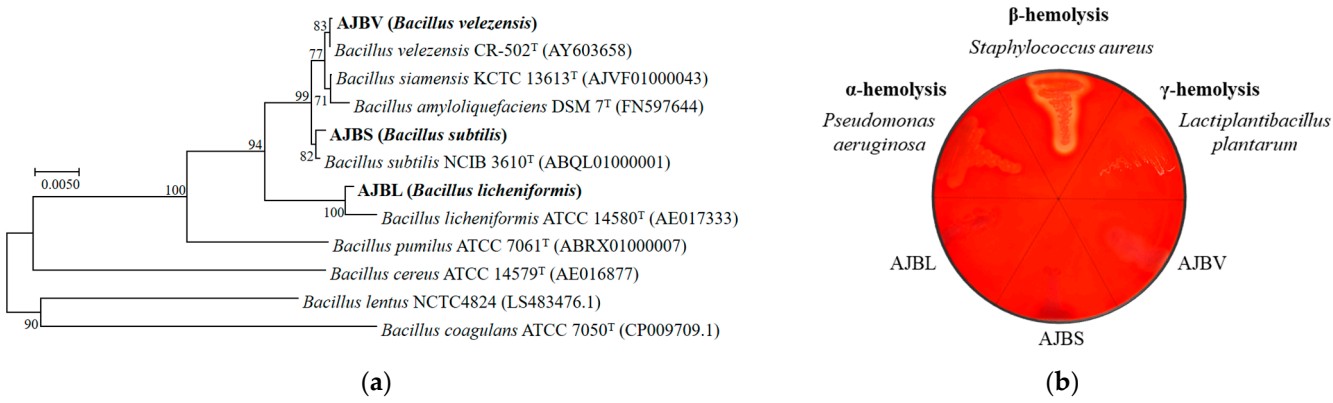

(**a**)    (**b**)

**Figure 1.** Phylogenetic tree and hemolytic activity analysis of the isolated strains. (**a**) The phylogenetic tree was constructed by the neighbor-joining method using Molecular Evolutionary Genetics Analysis version 7 (MEGA7) with 1000 bootstrap repetitions, after which the 16S rRNA sequences were aligned by the Clustal W program. NCBI accession numbers are presented in parentheses. (**b**) The hemolytic activity was measured using a blood agar plate, and *Pseudomonas aeruginosa* had α-hemolytic activity, *Staphylococcus aureus* had β-hemolytic activity, and *Lactiplantibacillus plantarum* having γ (non)-hemolytic activity were used as controls. AJBV, *Bacillus velezensis*; AJBS, *B. subtilis*; AJBL, *B. licheniformis*.

Previous studies have reported that the use of probiotics in aquaculture can provide beneficial effects on the growth and immunity of aquacultured animals [10]. Among many probiotic candidates, *Bacillus* can form spores that can survive in harsh environments such as low pH and high temperature, are non-pathogenic and non-toxic to the host, and produce various antibacterial substances [15]. As such, they have been evaluated as preferred candidates over other genera of probiotics [15]. The reason for selecting the *Bacillus* species as probiotic candidates for eel larvae in this study is because of their ability to improve

feed utilization, growth, and disease resistance. A variety of enzymes produced by *Bacillus* make them efficient symbionts to aid the metabolism of a wide range of lipids, proteins, and carbohydrates. Probiotics, therefore, may increase digestive enzyme activity that can contribute to fish growth [16,17]. In addition, the antibacterial substances they produce inhibit the growth of pathogenic microorganisms, thereby excluding pathogens through competition for nutrients and space, enhancing resistance to pathogens by upregulating the host's immunity, and subsequently increasing the survival rates of fish [18,19]. Therefore, in this study, three species of host-associated *Bacillus* were selected.

### 3.2. Growth and Survival Rate

The growth and survival rates of eel larvae supplemented with the artificial diet for 30 days are shown in Figure 2. The average total length (TL) of eel larvae was $10.44 \pm 0.87$ mm with no significant differences ($p > 0.05$) among the groups. Similarly, there were no significant differences in mean body depth (BD, average $1.15 \pm 0.09$ mm) and TL/BD values (average $11.11 \pm 1.23\%$). Regarding survival rates, the AJBV group ($24.00 \pm 12.16\%$) increased compared to controls ($14.90 \pm 1.27\%$), but no significant differences were observed. Survival rates of the AJBS and AJBL groups were $17.80 \pm 3.11\%$ and $14.10 \pm 1.56\%$, respectively, and there was no significant difference compared to the controls (Figure 2e).

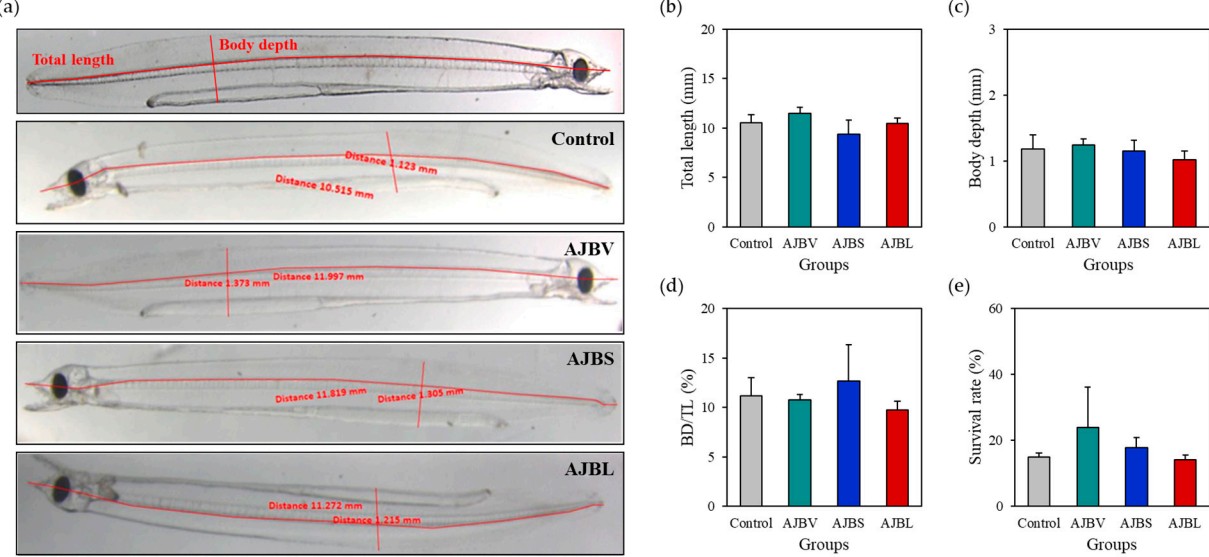

**Figure 2.** Growth and survival rate analysis of eel larvae fed a diet containing host-associated *Bacillus* species (AJBV, *Bacillus velezensis*; AJBS, *B. subtilis*; AJBL, *B. licheniformis*). (**a**) Total length and body depth were measured using the ZEN pro-2012 (blue edition) program on a stereo microscope. (**b**) Total length (TL), (**c**) body depth (BD), (**d**) BD/TL value, and (**e**) survival rate of eel larvae at 30 days after hatching are shown as group averages. The data are represented as the means $\pm$ standard deviation of three replicates (30 fish/replicate).

In a previous study using *Bacillus* as a probiotic in the aquaculture of olive flounder (*Paralichthys olivaceus*) [10,20,21], rockfish (*Sebastes schlegelii*) [13], red seabream (*Pagrus major*) [22,23], Nile tilapia (*Oreochromis niloticus*) [24], and whiteleg shrimp (*Litopenaeus vannamei*) [25], it was reported that fish growth and immunity increased. However, in this study, the three species of host-associated *Bacillus* did not affect the growth and survival rates of eel larvae. Although they did not provide beneficial effects on growth and survival rates, they also had no harmful effects. Perhaps under adverse conditions, *Bacillus* can interfere with the survival of pathogenic strains present in the body and breeding water and can prevent mass mortality due to pathogen infection. In this regard, further studies related to the antibacterial activity of the isolated strains, an extension of the feeding trial

period, and an analysis of the effect of using multiple strains need to be conducted to uncover the long-term benefits of probiotics and better understand how the microbiome is important.

### 3.3. Gene Expression Analysis

The gene expression profile of eel larvae fed a diet containing host-associated *Bacillus* at 30 days after hatching is shown in Figure 3. Compared to the controls, no significant differences were found in the relative expression levels of growth (growth hormone, growth hormone receptor 1, and insulin-like growth factor II-2) and digestive enzyme (amylase, trypsin, and lipase).

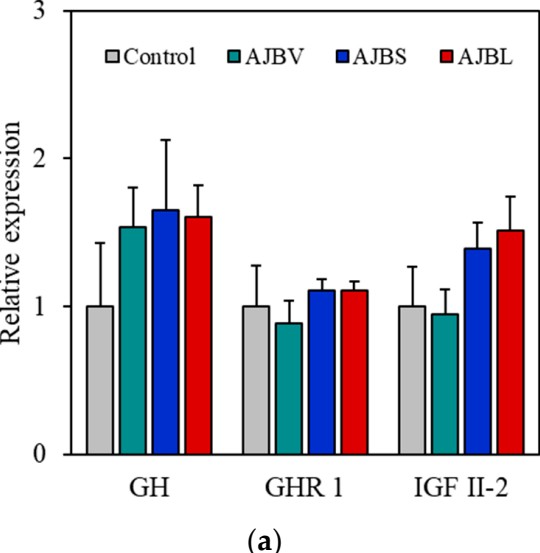

(**a**)

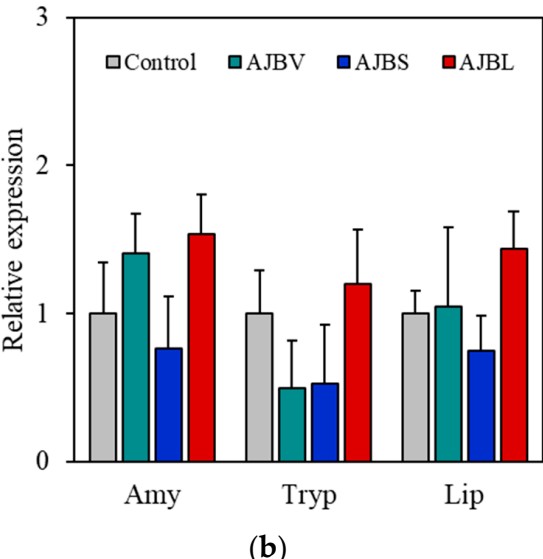

(**b**)

**Figure 3.** Gene expression profiles of eel larvae fed a diet containing host-associated *Bacillus* species (AJBV, *Bacillus velezensis*; AJBS, *B. subtilis*; AJBL, *B. licheniformis*). Comparison of gene expression 30 days after hatching was determined by RT-qPCR analysis. Levels of (**a**) growth (growth hormone, GH; growth hormone receptor 1, GHR 1; insulin-like growth factor II-2, IGF II-2) and (**b**) digestive enzyme (amylase, Amy; trypsin, Tryp; lipase, Lip) relative gene expression were quantified relative to β-actin transcription. The data are represented as the means ± standard deviation of three replicates (5 fish/replicate).

Many previous studies have reported that supplementation of probiotics in feed contributes to increased expression of growth-related hormones and digestive enzyme genes in farmed fish [26,27]. However, in this study, supplementation with the three strains of host-associated *Bacillus* did not significantly affect eel larval growth and digestive enzyme gene expression. The development of the digestive system of eel larvae is very delayed compared to other fish larvae [28]. Their stomach develops during metamorphosis (269–311 days after hatching), meaning that food ingested during the early larval stage enters the intestine directly [29,30]. Ingested food does not remain in the intestine long enough for digestion and absorption of nutrients and is excreted immediately. In addition, the supplemented host-associated *Bacillus* cannot colonize the host's intestines because they are excreted together. For this reason, short-term host-associated *Bacillus* supplementation may not be effective for eel larvae. However, supplying fermented feed using host-associated *Bacillus* to replace the digestive process in the stomach or supplementing the diet with host-associated *Bacillus* long after the stomach has developed (after 311 days) may have a positive effect on eel growth and survival rates.

## 4. Conclusions

In conclusion, feed supplementation with host-associated *Bacillus* did not significantly affect the growth and survival rates of eel larvae reared on an artificial diet in the early stages of life. Further research will evaluate the growth and survival rate of eel larvae by supplying fermented feed using the three species of host-associated *Bacillus* isolated in this study.

**Author Contributions:** Conceptualization, W.J.J., S.-K.K. and J.M.L.; methodology, W.J.J., S.-K.K., S.Y.P., D.P.K., Y.-J.H., H.K., S.-J.L. and M.G.S.; validation, W.J.J., S.-K.K., S.Y.P., D.P.K., Y.-J.H., H.K. and S.-J.L.; investigation, S.Y.P., D.P.K., Y.-J.H., H.K. and S.-J.L.; writing—original draft preparation, W.J.J. and J.M.L.; writing—review and editing, W.J.J. and J.M.L.; supervision, E.-W.L., S.L. and J.M.L.; project administration, E.-W.L., S.L. and J.M.L.; funding acquisition, E.-W.L., S.L. and J.M.L. All authors have read and agreed to the published version of the manuscript.

**Funding:** This research was funded by a grant from the National Institute of Fisheries Science, Republic of Korea (R2023026).

**Institutional Review Board Statement:** All experiments were performed following the guidelines of the Korean Association for Laboratory Animals (approval no. 2022-NIFS-IACUC-8).

**Data Availability Statement:** The datasets generated during and/or analyzed during the current study are available from the corresponding author on reasonable request.

**Conflicts of Interest:** The authors declare no conflict of interest.

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
