# Peer review of "Effect of Host-Associated Bacillus-Supplemented Artificial Diets on Growth, Survival Rate, and Gene Expression in Early-Stage Eel Larvae (Anguilla japonica)"

_fishes, doi:10.3390/fishes8050247_

Round 1
Reviewer 1 Report
In this manuscript entitled “Effect of Host-Associated Bacillus Supplemented Artificial Diets on Growth Survival Rate, and Gene Expression in Early-Stage Eel Larvae (Anguilla japonica)”, authors investigated the effect of host-associated microorganisms on the growth rate, survival rate, and gene expression of eel larvae. Authors reveled that host-associated bacillus-supplemented diets do not affect the growth, survival rates and gene expression of eel larvae.
There are no serious technical problems, although there is no positive effect on the new diet for eel larvae. Thus, this article may be worth of publishing. However, several corrections may be necessary according to the following suggestions.
1. Some of the references are inappropriate.
e.g.
“the natural stock of Japanese eels has noticeably declined over the past few decades [1, 5]” Kagawa and Tanaka 2005 and Shi et al. 2020 are not research papers on eel resources.
“Therefore, artificial propagation…, which can stabilize eel prices [3]” Jang et al. is not appropriate in this sentence.
“Research on the production…. , but also in Korea and China [6, 7]” Hibiya 1970 and Yamamoto and Yamauchi 1974 are study of Japanese eel. You should add Korean and Chinese articles in this sentence.
2. There are subtypes of growth hormone receptors and IGFs. You should describe which subtype you measured.
3. “Perhaps under adverse conditions, /// to pathogen infection.” In this study, the survival rate is relatively low (<20%). Thus, I think the rearing condition is already adverse.
4. “However, supplying fermented feed /// eel growth and survival rates”. In anguillid eels, stomach developed during metamorphosis, and during metamorphosis they does not eat. Thus, this statement is contradictory.
Author Response
We greatly appreciate to the Reviewer for evaluating our manuscript. We tried to respond to the honorable reviewer comments point-by-point, indicating necessary changes in the revised version. We believe these changes have substantially improved the quality and clarity of this article. Please refer to the attached file for replies to comments.

Reviewer 2 Report
The objective of the study was “Effect of Host-Associated Bacillus-Supplemented Artificial Diets on Growth, Survival Rate, and Gene Expression in Early-Stage Eel Larvae (Anguilla japonica)”. Indeed a good objective is chosen for the improvement of the aquaculture of this fish. Overall, the manuscript presents generally a sound study. There are shortcomings with some of the practical aspects of the work and with the structure of the manuscript and the way in which the material and methods has been presented. The discussion section especially is highly speculative and needs to more tightly focus on the results and implications. I have provided specific comments about my concerns regarding the novelty and the importance of this study as they arise in the manuscript. The manuscript should be accepted after minor revision.
ABSTRACT
The abstract of this article looks ok. The abstract section should briefly introduce the research background and significance, clarify the research methods, introduce the main research results, and finally, give the corresponding conclusions.
INTRODUCTION
The introduction section is very well written and needs no modification.
MATERIALS AND METHODS
The presentation of the M & Ms varies in detail with insufficient information being given about some of the experimental protocols and sampling, and the presentation is disjointed and confusing. This section need to be re-integrated and checked to avoid confusion. All experimental protocols should be better explained. There is a paucity of detail given about the sampling protocols.
There was no description on the culture conditions during the acclimation period? The authors mentioned in the section 2.3 that Eel larvae were provided by the National Institute of Fisheries Science. Was the larvae acclimated to the laboratory condition?
How larvae were transported? Do authors check any infection and simultaneously gave any prophylactic dip treatment? How did authors know that larvae were healthy?
For the feeding trial, was this feeding trial carried indoor or outdoor? What was the water source?
How the water quality had been measured or monitored?
The flow rate of water is 0.99–1.13 L/min? Was it enough? And why other important water quality parameters like C02, alkalinity was not determined?
The authors should have provided the diet preparation in more detail.
In what form the diet was fed to larvae?
Section 2.5: elaborate it and more importantly authors should have gone for western blotting.
The authors need to make sure that all figures should be self-explanatory.
In my opinion that authors should reframe the manuscript as per suggestions and quarries raised. Presentation and language needs to be improved throughout the manuscript. The manuscript should be accepted after minor revision.
Author Response
We have received your considerate review of the manuscript. It was a great pleasure to see your insightful and helpful comments. We hope our responds to your comments finds well in addressing most of the mentioned inquiries. Please refer to the attached file for replies to comments.

Round 2
Reviewer 1 Report
The authors should correct the points listed below.
Inappropriate citations. You cite the introduction of the paper, but in general you should cite the results of the paper.
You measured mRNA expression of GHR1. You should write "GHR1". The same is true for IGF.
Author Response
We greatly appreciate to the Reviewer for evaluating our manuscript. We tried to respond to the honorable reviewer comments point-by-point, indicating necessary changes in the revised version. We believe these changes have substantially improved the quality and clarity of this article.
We referred to various papers to explain the basic background knowledge in the introduction. Most of the similar contents were confirmed, such as the decrease in the amount of capture of wild eel larvae, the increase in the price of eel, and the emphasis on the need to develop an eel breeding system, and we referred to several representative papers.
Throughout the paper, including Figure 3a, the names of GHR and IGF were changed to GHR 1 and IGF II-2, respectively.
We thank you again for your consideration of this manuscript. Please do not hesitate to contact me directly if necessary.
Yours sincerely,
Authors,